# Omega-3 Polyunsaturated Fatty Acids Provoke Apoptosis in Hepatocellular Carcinoma through Knocking Down the STAT3 Activated Signaling Pathway: In Vivo and In Vitro Study

**DOI:** 10.3390/molecules27093032

**Published:** 2022-05-09

**Authors:** Noura M. Darwish, Mohamed M. A. Elshaer, Saeedah Musaed Almutairi, Tse-Wei Chen, Mohamed Othman Mohamed, Wael B. A. Ghaly, Rabab Ahmed Rasheed

**Affiliations:** 1Department of Biochemistry, Faculty of Science, Ain Shams University, Cairo 11566, Egypt; 2Ministry of Health Laboratories, Tanta 16335, Egypt; 3Department of Clinical Pharmacology, Faculty of Medicine, Ain Shams University, Cairo 11566, Egypt; mohamed_elshaer@med.asu.edu.eg; 4Department of Clinical Pharmacology, Faculty of Medicine, King Salman International University, South Sinai 46511, Egypt; 5Department of Botany and Microbiology, College of Science, King Saud University, Riyadh 11451, Saudi Arabia; alsaeedah@ksu.edu.sa; 6Department of Materials, Imperial College London, London SW7 2AZ, UK; twchenchem@gmail.com; 7Anatomy Department, Faculty of Medicine, King Salman International University, South Sinai 46511, Egypt; mohamed.othman@ksiu.edu.eg; 8Physiology Department, Faculty of Medicine, Fayoum University, Fayoum 63511, Egypt; wba00@fayoum.edu.eg; 9Physiology Department, Faculty of Medicine, King Salman International University, South Sinai 46511, Egypt; 10Histology & Cell Biology Department, Faculty of Medicine, King Salman International University, South Sinai 46511, Egypt

**Keywords:** n-3 polyunsaturated fatty acids (PUFAs), apoptosis, STAT3, cyclin D1, Bcl-2, HepG2

## Abstract

Hepatocellular carcinoma (HCC) is a common type of liver cancer and is a leading cause of death worldwide. Signal transducer and activator of transcription 3 (STAT3) is involved in HCC progression, migration, and suppression of apoptosis. This study investigates the apoptotic effect of the dietary antioxidant (n-3 PUFAs) on HepG2 cells and analyzes the underlying molecular mechanisms of this effect both in vivo and in vitro. In vivo study: Seventy-five adult male albino rats were divided into three groups (n = 25): Group I (control): 0.9% normal saline, intraperitoneal. Group II: N-Nitrosodiethylamine (200 mg/kg b.wt) intraperitoneal, followed by phenobarbital 0.05% in drinking water. Group III: as group II followed by n-3 PUFAs intubation (400 mg/kg/day). In vivo study: liver specimens for biochemical, histopathological, and immunohistochemical examination. In vitro study: MTT assay, cell morphology, PCR, Western blot, and immunohistochemical analysis. n-3 PUFAs significantly improved the histopathologic features of HCC and decreased the expression of anti-apoptotic proteins. Further, HepG2 cells proliferation was suppressed through inhibition of the STAT3 signaling pathway, cyclin D1, and Bcl-2 activity. Here we report that n-3 PUFAs may be an ideal cancer chemo-preventive candidate by targeting STAT3 signaling, which is involved in cell proliferation and apoptosis.

## 1. Introduction

Liver cancers are ranked fourth among all cancer types worldwide [1,2]. Hepatocellular carcinoma (HCC) is the most common type (85–90%) of primary liver cancer, representing a real challenge and a heavy burden on health care systems in various countries around the world. Diverse risk factors are involved in the pathogenesis of HCC. Hepatitis B virus infection remains the main risk factor, followed by alcohol consumption, Hepatitis C virus, non-alcoholic steatohepatitis, ingestion of Aflatoxin B1, and hemochromatosis [3,4,5,6,7,8]. These factors initiate liver tumor, an irreversible genetic mutation with subsequent proto-oncogene activation [6]. For tumor development, initiation must be followed by a promotion process that requires inflammation as an important factor [9,10]. Risk factors and subsequent clinical outcomes vary depending on geographical distribution, genetic diversity, and molecular subtypes [3,4,5,6]. The HCC shows a preponderance (85% of diagnosed cases) in developing countries in Africa and Eastern Asia [9,10,11,12]. The average 5-year survival rate in HCC patients is unfavorably low, ranging from 5% to 14% [13]. This could be attributed to late diagnosis, recurrence, and the tendency of HCC cells to metastasize [14]. It is impossible to improve therapeutic options for HCC without significant knowledge of the above-stated mechanisms.

Signal transducer and activator of transcription (STAT family) are mediator proteins that transfer signals from the extracellular matrix (ECM) into the nucleus [15,16]. Currently, seven mediators of STAT family proteins are recognized [17]. The STAT3, for example, is a transcription mediator that regulates cell growth and apoptosis in normal living cells [17,18]. In several studies, STAT3 has significantly impacted cancer progression and related genes [19].

The STAT3 signaling is frequently activated in some solid cancers, including HCC and multiple myeloma [20,21,22,23]. In addition, STAT3 activation is associated with epithelial-mesenchymal tumors [EMT] and plays a critical role in tumorigenicity, anti-apoptosis, prolonging the survival of malignant cells, and metastasis [7,20,21,24,25].

The binding of STAT3 activators (IL-6 and other growth factors) to their corresponding receptors on hepatocytes induces Janus kinase (JAK) or Src phosphorylation and, subsequently, STAT3 phosphorylation. Phosphorylated STAT3 translocates into the nuclei to induce transcription of genes involved in cell survival and proliferation, including cyclin D1 [7,8], as shown in (Figure 1). Cyclin D1 controls cell cycle progression, and its overexpression is associated with many cancers [26].

Knocking down of p-STAT3 signaling could induce apoptosis and down-regulate matrix metalloproteinases, thus, suppressing cancer cell invasion [27]. This reveals that the STAT3 pathway is of substantial importance in cancer chemoprevention and could be a promising clinical target for liver cancer patients. Thus, intense efforts have been made to identify suitable anti-STAT3 agents to treat human cancers such as HCC [28].

Many dietary antioxidants have potent anti-tumor activity targeting the STAT3 signaling network [29,30]. In recent years, the popularity of polyunsaturated fatty acids (PUFAs) has increased because of their numerous functions in health promotion; minimum side effects; and decreasing risk of diseases such as cancer, cardiovascular diseases, diabetes, and obesity [31,32]. Many clinical trials have been conducted to demonstrate the efficacy of n-3 PUFAs in cancer prevention and treatment in experimental animals and cell culture [33]. Previously, it was reported that increased levels of n-3 PUFAs in tissues significantly repressed liver carcinogenesis [34]. In human HCC cell lines, dose-dependent n-3 PUFAs, EPA, and DHA treatment reduced cell viability via caspase-3 and caspase-9 cleavage [32]. Although many molecular targets of n-3 PUFAs have been identified and multiple mechanisms appear to underlie the induction of apoptosis, the mechanisms are still not completely understood [31,35]. The current study aimed to examine the potential chemoprotective (in vivo) and antiproliferative (in vitro) effects of n-3 PUFAs on diethyl-nitrosamine (DEN)-induced HCC and human hepatocellular carcinoma cells (HepG2), respectively.

In our study, we noticed that n-3 PUFAs markedly improved the histopathological picture of DEN-induced hepatocytes’ atypia and decreased the expression of antiapoptotic proteins. Further, we reported the growth-inhibitory effect of n-3 PUFAs on cancer HepG2 cells by apoptosis induction and cell cycle arrest. We particularly evaluated the n-3 PUFAs-induced changes in the expression of pSTAT3 and their effects on the expression of the proliferative protein cyclin D1 that controls the cell cycle and the anti-apoptotic Bcl-2 protein. Our results showed that n-3 PUFAs could be a potential adjuvant therapy for HCC.

## 2. Results

### 2.1. Effect of DEN And/Or n-3 PUFAs on Serum Levels of Liver Function Markers

The outcomes are shown in Table 1 as mean ± standard deviation (SD). We noticed a significant increase (*p* < 0.05) in the activities of alanine aminotransferase (ALT) and aspartate transaminase (AST) in the DEN group compared to the control group. In contrast, no significant changes were observed in serum alkaline phosphatase (ALP), gamma-glutamyl transferase (GGT), and albumin (ALB) concentrations in both the DEN group and the n-3 PUFAs group compared to the control group. A significant reduction (*p* < 0.05) in ALT and AST activities was marked in the n-3 PUFAs group compared to the DEN group (Table 1).

### 2.2. Effect of DEN And/Or n-3 PUFAs on the Histopathological Picture of Liver Tissue

Histopathological examination of hematoxylin and eosin (H&E)-stained sections of liver tissue of the control group showed normal hepatic architecture formed of cords of acidophilic hepatocytes with uniform nuclei radiating from the central vein, separated by blood sinusoids (Figure 2A). In the DEN group, the liver tissue lost its typical architecture. Hepatocytes showed atypia, cellular and nuclear pleomorphism, hyperchromasia with distinct nucleoli, and markedly vacuolated cytoplasm. Large nodules with focal necrosis and hyalinosis surrounded by marked inflammatory infiltrates were observed. At some points, there was a pseudo-glandular pattern of hepatocytes. Hepatic sinusoids and central vein were dilated (Figure 2B,C). n-3 PUFAs group revealed nearly normal liver architecture with regular hepatocytes’ cords except for mildly dilated sinusoids and congested central vein (Figure 2D).

### 2.3. Effect of DEN And/Or n-3 PUFAs on Immunohistochemical Results

The outcomes of morphometric measurements are presented in Appendix A as mean ± SD. Examination of immune-stained sections of different research groups revealed strong positive expression of p-STAT3, cyclin D1, and Bcl-2 in the DEN group with a highly significant increase (*p* < 0.001) compared to the control group. On the other hand, p-STAT3, cyclin D1, and Bcl-2 expression showed a highly significant regression (*p* < 0.001) in the n-3 PUFAs group as compared to the DEN group (Figure 3).

### 2.4. Effect of n-3 PUFAs on DNA Fragmentation

A degree of hepatic DNA fragmentation was observed in the n-3 PUFAs group. At the same time, there was no fragmentation observed in the control group, indicating the role of n-3 PUFAs in the assessment of DNA damage (Figure 4 and Appendix A).

### 2.5. Effect of n-3 PUFAs on HepG2 Cell Viability

HepG2 cells exhibited approximately the same viability percentage at concentrations of 0–600 μM (*p* > 0.05). n-3 PUFAs significantly decreased the viability of cells compared with the control group at concentrations of 900 μM (*p* < 0.05) and maintained the same activity of cell viability at higher concentrations. Notably, at a concentration of 900 μM, n-3 PUFAs exhibited one-half percentage (IC_50_) of the control group cell viability with a significant (*p* < 0.05) decrease in cell viability (Figure 5). Interestingly, n-3 PUFAs showed significant antiproliferative activity against HepG2 cell growth at 800 μM concentration (lower than IC_50_) (Figure 5).

### 2.6. Effect of n-3 PUFAs on Morphologic Observations

Good apoptotic features of treated HepG2 cells with (900 µM), including the presence of spindle-shaped cells and cell shrinkage, were observed. Slight apoptotic morphological changes were observed at half IC_50_ (450 µM). Untreated HepG2 cells showed high confluency of monolayer cells (Figure 6 and Appendix AA–C).

### 2.7. Effect of n-3 PUFAs on DNA Fragmentation Assay

DNA laddering indicates the presence of double-strand breaks. Treated HepG2 cells at IC_50_ concentration (900 µM) showed the typical formation of DNA fragments as ladders (Figure 7 and Appendix A4). No fragments were detected in cells incubated with one-half IC_50_ and in the absence of n-3 PUFAs.

### 2.8. Effect of n-3 PUFAs on p-STAT3 and Related Candidate Genes’ Expression

A significant (*p* ˂ 0.05) decrease in the expression level of p-STAT3, cyclin D1, and Bcl-2 was shown after 24 h treatment with 900 µM n-3 PUFAs compared to untreated HepG2 cells. In contrast, no significant (*p* > 0.05) change in the expression levels of p-STAT3 was observed after treatment with half IC_50_ (450 µM) when compared to untreated HepG2 cells (Figure 8).

### 2.9. Effect of n-3 PUFAs on p-STAT3 and Related Proteins in HepG2 Cells

Western blot analysis for STAT3, p-STAT3, Cyclin D1, and Bcl-2 showed no change in the expression level of all proteins when observed under IC_25_ (450 μM) treatment. In comparison, the Bcl-2 expression level was decreased under the effect of both IC_25_ (450μM) and IC_50_ (900 µM) 24 h treatment, whereas there was decreased expression of p-STAT3 and Cyclin D1 proteins after treatment with IC_50_ (900 µM) of n-3 PUFAs (Figure 9 and Appendix A).

## 3. Discussion

HCC is a primary liver malignancy and is a leading cause of cancer-related death worldwide [10]. HCC is characterized by a lack of early diagnosis, so novel preventive and therapeutic approaches are definitely needed [3,4,36]. Recent experiments have been directed toward elucidating and understanding the signaling networks involved in HCC progression as potential therapeutic targets [5]. The STAT3 is of importance due to its role in enhancing cancer progression, malignancy, cancer cell migration, and suppression of apoptosis [37]. Activated STAT3 has been shown to have anti-apoptotic activity and enhance the survival of malignant cells in HCC by activating the expression of anti-apoptotic protein Bcl-xL and Bcl-2 [38,39,40,41,42,43]. Furthermore, active STAT3 binds to human cyclin D1 promoters and raises cyclin D1, which is connected to cancer progression and development. [44]. Therefore, STAT3 molecular signaling pathways can be considered a downstream target for cancer therapy.

Some natural products are capable of targeting the STAT3 signaling pathway and suppressing malignancy and proliferation [15]. n-3 PUFAs are dietary components that are involved in a variety of physiological processes and have a potential preventive effect on some diseases, including cancer. Several studies have shown that diets containing n-3 PUFAs retarded the growth and metastasis of primary tumors in humans, such as liver, breast, prostate, and colon cancers [45,46,47].

Moreover, previous studies reported that supplementation with n-3 PUFAs could induce apoptosis and reduce cell proliferation in cultured cancer cells [48,49,50,51].

In the present study, the histopathological examination of the liver of the DEN-treated group showed lost architecture. Hepatocytes showed atypia, cellular, and nuclear pleomorphism, with hyperchromasia and distinct nucleoli, and a markedly vacuolated cytoplasm. Large nodules with focal necrosis and hyalinosis surrounded by marked inflammatory infiltrates were observed. At some points, there was a pseudo-glandular pattern of hepatocytes. Hepatic sinusoids and central veins were dilated.

These findings coincide with a recent study that reported similar lesions in rats treated with DEN in the form of dysplastic hepatocytes with large nuclei and conspicuous macronucleoli, mild mononuclear inflammatory infiltrate, fibrosis, lytic necrosis, frequent apoptotic bodies, and distended portal veins [52]. Alongside, Kerdput et al. [53] stated that DEN-treated rats showed a pseudo-glandular pattern of hepatocytes with hepatic cords greater than three cell layers in thickness and intracellular hyaline globules. In addition, areas of steatosis and micro-steatosis marked by cytoplasmic lipid vacuoles were mentioned by Alsahli et al. [54] and Nunes et al. [55]. In the same context, a previous study affirmed that DEN-induced liver injury was severe and showed dilated congested blood vessels, edema, and inflammatory cellular infiltrates [54].

Interestingly, in the current study, nearly normal liver architecture with regular hepatocytes was observed in the n-3 PUFAs-treated group, except for mildly dilated sinusoids and congested central vein, indicating the protective effect of n-3 PUFAs. These results come in accordance with previous results, which reported reduced tumorigenesis and enhanced survival rate upon increased n-3 PUFAs tissue levels for DEN-induced liver tumors in mice in a rat multiorgan cancer model [56,57].

Immunohistochemical analyses of p-STAT3, cyclin D1, and Bcl-2 showed a strong positive expression in the liver of DEN-administered rats compared to the control group, whereas n-3 PUFAs-treated rats showed a marked reduction of these antiapoptotic proteins compared to the DEN group, confirmed by the morphometric measurements and the statistical analysis.

These findings harmonize with a prior study that reported the upregulation of STAT3, its phosphorylated form (p-STAT3), and its downstream target genes cyclin D1 and Bcl-2 in HCC compared to normal liver cells, which play a critical role in cancer liver promotion and metastasis [43,58].

Similar to our results, Zhang et al. [59] proved that the levels of p-STAT3, Cyclin D1, and Bcl-2 showed a significant down-regulation in cancer liver cells through knocking down the expression of STAT3 by adenovirus infection. Therefore, it is of note that the oral administration of n-3 PUFAs may suppress the occurrence of HCC by the exact mechanism.

In the present study, n-3 PUFAs inhibited HepG2 cell proliferation and caused a significant reduction in viable cell number after treatment with IC_50_ concentration (900 µM). Further, HepG2 cells treated with n-3 PUFAs showed morphological changes characterized by apoptotic body formation and DNA fragmentation. This could be a potential explanation for anti-cancer activity induced by n-3 PUFAs, suggesting that the apoptotic action of n-3 PUFAs is dependent on the activation of apoptosis.

For further elucidation of the inhibitory effect of n-3 PUFAs on HepG2, the mechanism of action on cells signaling was examined by RT-PCR and Western blot analysis for p-STAT3 and its related mediators (cyclin D1 and Bcl-2). In our study, a significant downregulation of p-STAT3, Bcl-2, and Cyclin D1 genes’ expression was observed after treatment at IC_50_ concentration for 24 h. Western blot analysis results demonstrated decreased levels of p-STAT3, cyclin D1, and Bcl-2 proteins in n-3 PUFAs-treated HepG2 cells compared with their expression in untreated cell extract. Overall, recent findings imply that the n-3 PUFAs’ growth-inhibitory action on HepG2 is linked to induced suppression of p-STAT3, cyclin D1, and Bcl-2 expression.

The results of this study are consistent with previous studies that reported the antitumor actions of n-3 PUFAs [51,60].

The n-3 PUFAs inhibit growth signal transduction and down-regulate some genes, which are signaling mediators and often elevated in carcinogenesis [61].

Therefore, further examination of the anti-hepatocarcinogenic effects of n-3 PUFAs employing the DEN-induced hepatocarcinogenic rat model is established in this study.

The oral administration of n-3 PUFAs to DEN-induced hepatocarcinogenic rats resulted in significantly lower serum transaminase levels and tumor formation in the liver. It was previously reported that fish consumption is independently correlated with a reduced risk of HCC [62].

## 4. Materials and Methods

### 4.1. Ethical Statement

We conducted this study under NIH guidelines for the Care and Use of Laboratory Animals. The practices were conventional and approved by the Animal Care and Use Committee, Faculty of Pharmacy—Ain Shams University, Cairo, Egypt (approval number ACUC-FP-ASU RHDIRB2020110301 REC# 56).

### 4.2. Chemicals

DEN in a liquid form [25 mL in a glass bottle, product No. N0756], phenobarbital [PB] in a powder form [pack size 100 gm, product No. P1636], and Omega-3 (n-3 PUFAs) as free fatty acids in the form of pure fish oil [pack size 100 mg, product No. 47085-U] were all purchased from Sigma Aldrich Chemical Co. (St. Louis, MO, USA). The chemical structures of DEN, PB, and n-3 PUFAs are shown in Appendix A. n-3 PUFAs were diluted in ethanol at a stock concentration of 0.5 M. The stock was aliquoted in dark-colored glass vials with screw tops (Agilent Technologies, Wokingham, UK) to be protected from light and oxidation and stored at −20 °C. All other chemicals and biochemicals used in our experiment were of analytical grade. Biochemical kits for serum analysis were purchased from the Gamma Trade Company for Pharmaceutical and Chemicals, Dokki, Egypt.

### 4.3. Antibodies

Monoclonal antibodies to β-actin IgG, STAT3, p-STAT3, cyclin D1, and Bcl-2 (Santa Cruz Biotechnology, Paso Robles, CA, USA) were diluted at 1:500. Secondary antibodies conjugated with horseradish peroxidase (HP) were diluted (1:5000) (Santa Cruz Biotechnology).

### 4.4. In Vivo Study

#### 4.4.1. Sample Size Calculation

The study aimed to investigate the impact of adding n-3 PUFAs on DEN-induced HCC in rats when administered orally with a daily dose of 400 mg/kg. According to Stevenson mark [63], assume the true mean cure rate of the treatment is 0.85. We consider a difference of <0.10 in cure rate to be of no importance (delta = −0.10). Assuming a one-sided test size of 5% and a power of 80%,
(1)$$\frac{N\;\left(B\right)=P\left(A\right)(1-P\;\left(A\right)+P\left(B\right)(1-P\;\left(B\right)\;\left[\;Z\;1-\left(\alpha /2\right)+Z\;1-\beta \right]2}{\;r\;P\left(A\right)-P\left(B\right)-\delta }$$
so, the number of animals in each arm N = (0.355/1) (2.48/0.3)^2^ = 25 rats.

#### 4.4.2. Experimental Animals

Seventy-five Sprague–Dawley male rats (weighing 100–150 gm, 10–12 weeks) were purchased from the Animal House, Faculty of Pharmacy, Ain Shams University, Cairo, Egypt. Animals were housed in stainless-steel cages (five rats/cage) and maintained under a 12 h:12 h light/dark cycle with constant temperature (22–24 °C) and humidity. Animals were allowed free access to a standard diet and tap water. One week previous to the experimental regimen, the animals were permitted to acclimate.

#### 4.4.3. Experimental Design

For a total duration of 36 weeks, rats were used to study the protective effect of n-3 PUFAs against DEN-initiated/PB-promoted hepatocellular carcinoma. Rats were equally and indiscriminately divided into three groups (n = 25): Group I (Control): single intraperitoneal injection of 0.9% normal saline solution. Group II (DEN): single intraperitoneal injection of freshly prepared DEN in normal saline at a dose of 200 mg/kg body weight. Two weeks later, PB (0.05%) was added to the drinking water for 24 consecutive weeks [64]. Group III (n-3 PUFAs): the same as group II followed by intragastric intubation of n-3 PUFAs dissolved in propylene glycol at a dose of 400 mg/kg b.wt/day for a further 10 weeks [65]. At the end of the experiment, all rats were sacrificed by cervical dislocation after an overnight fast. The timeline of this study is presented in Figure 10.

#### 4.4.4. Determination of Biochemical Parameters

Blood samples were collected from the tail vein of the animals just before sacrificing, and the serum levels of ALT, AST, ALP, ALB, and GGT were tested using an automatic analyzer (Biosystem, Muttenz, Switzerland).

#### 4.4.5. Histopathology

Based on Bancroft and Gamble [66], after sacrificing, liver specimens from all rats of various research groups were fixed in 10% formol saline, further processed for paraffin sections (4 μm thick), and stained with H&E. The sections were examined and photographed using ordinary light microscopy (Olympus, Tokyo, Japan) to report the histopathological changes.

#### 4.4.6. Immunohistochemistry and Morphometric Studies

Immunostaining of p-STAT3, cyclin D1, and Bcl-2 was carried out using a standard protocol [67]. Tissue sections were incubated with primary anti-p-STAT3, anti-cyclin D1, and anti-Bcl-2 antibodies (each at 1:100 dilutions, Santa Cruz Biotechnology) at 4 °C overnight. Sections were then counterstained with hematoxylin. The image analysis system Leica Qwin DW3000 (LEICA Imaging Systems Ltd., Cambridge, UK) was used to snap the most representative 10 non-overlapping random fields to measure the mean area % of p-STAT3, cyclin D1, and Bcl-2 expression at magnification ×400 in all immune-stained sections in all groups. Measurements were performed and gathered in the Department of Histology and Cell Biology labs, Faculty of Medicine, King Salman International University, El Tor, South Sinai, Egypt.

### 4.5. In Vitro Study

#### 4.5.1. Cell Culture

HepG2 cells (ATCC) were seeded in Dulbecco’s Modified Eagle Medium (DMEM) containing L-glutamine, with high glucose, sodium bicarbonate, 1% penicillin/streptomycin, and 10% Fetal Bovine Serum (GIBCO, Waltham, MA, USA), at 5% CO_2_ for 24 h at 37 °C. Cells were grown to 60% confluence, trypsinized with 0.25% trypsin/2 mM EDTA, and sub-cultured for experimental use [68].

#### 4.5.2. MTT Assay

Cells were cultured in a 96-well plate at a density of 2.0 × 104 cells/well, as previously described, and were treated with n-3 PUFAs at concentrations ranging from 400 μM to 1000 μM and incubated at 37 °C in 5% CO_2_ for 48 h, followed by incubation with 5 mg/L MTT for 4 h. Dimethyl sulfoxide (DMSO) was added (100 μL), and absorbance at 550 nm was measured with a microplate reader (Model APW-100; BioTek, Hangzhou, China). The inhibitory concentration (50% binding) value (IC_50_) was obtained from the concentration-effect curves [69].

#### 4.5.3. Morphological Changes Observation

The morphological changes of all the cells were observed under inverted light microscope at ×40 magnification after treatment of cells with n-3 PUFAs at IC_50_ (900 µM) and IC_25_ (450 µM) for 48 h.

#### 4.5.4. DNA Fragmentation

Liver tissues homogenate was incubated in 100 mM Tris–HCl (pH 8.0), 25 mM EDTA, 0.5% SDS, and 0.1 μg/mL proteinase K at 60 °C for 3 h. HepG2 cells were pelleted after treatment with n-3 PUFAs at IC50 and IC25 concentration and incubated for 48 h. Untreated cells were used as a negative control. Both types of cells (liver tissue and HepG2 cells) were used to extract DNA according to a previous protocol [70]. The extracted DNA was analyzed by electrophoresis on a 1.5% agarose gel. DNA fragments were visualized on the gel stained with ethidium bromide under UV light.

#### 4.5.5. Real-Time Reverse Transcription Polymerase Chain Reaction

A total of 2.0 × 105 cells/well were seeded in 24-well plates treated with n-3 PUFAs for 24 h. RNA was extracted by TRIzol reagent. RNA samples were quantified using NanoDrop spectrophotometer ND 2000 (Thermo Fisher Scientific, Waltham, MA, USA). cDNAs were synthesized from RNA using M-MLV reverse transcriptase (GeNeiTM, Bangalore, India). Glyceraldehyde-3-phosphate dehydrogenase was considered the reference gene. The reaction was performed on a Rotor-gene 6000 System (Corbett Research, Mortlake, New South Wales, Australia) in 20 µL Master Mix (Qiagen, Germantown, MD, USA). The reaction was performed under the following conditions: 95 °C for 15 min; 45 cycles of 94 °C for 15 s, 55 °C for 40 s, and 72 °C for 20 s. The expression level under the effect of n-3 PUFAs was also calculated in terms of relative fold change. The primer sequences were as follow: GAPDH: forward: 5′-CCACTCCTCCACCTTTGAC-3′, Reverse: 3′-ACCCTGTTGCTGTAGCCA-5′; BCl2: forward: 5′-CTTTGAGTTCGGTGGGGTCA-3′, Reverse: 3′-GGGCCGTACAGTTCCACAAA-5′; Cyclin D1: forward: 5′-AGGAACAGAAGTGCGAGGAG-3′; reverse, 3′-CACAGAGGGCAACGAAGGT-5′; STAT3: forward: 5′-CTTTGAGACCGAGGTGTATCACC-3′, Reverse: 3′-GGTCAGCATGTTGTACCACAGG-5′.

#### 4.5.6. Western Blotting

Cells were plated at 1.5 × 104 cell/ well into 6-well plates and treated with n-3 PUFAs at concentrations of 900 and 450 µM, respectively. Samples were homogenized in radioimmunoprecipitation assay buffer supplemented with a protease inhibitor cocktail and a phosphatase inhibitor cocktail (Sigma-Aldrich, Burlington, MA, USA) and were kept on ice for 30 min. Samples were analyzed with Western blotting with 12% resolving sodium dodecyl sulfate-polyacrylamide gel electrophoresis and transferred onto a polyvinylidene fluoride membrane. Non-specific sites were blocked with skimmed milk for 2 hours; the membrane was incubated with specific primary antibodies (1:500; Cell Signaling, Danvers, MA, USA) at room temperature, washed three times for 15 min with Tris-buffered saline containing 0.1% Tween-20, and further incubated with goat anti-rabbit IgG-HRP for another 1 h. β-actin was used as a positive control using β-actin monoclonal antibody (Sigma) used at 1:500 [71].

### 4.6. Statistical Analysis

A summary of the outcomes was given as mean ± standard deviation (SD). The statistical analysis was performed using the Statistical Package of Social Science (SPSS, Version 22, IBM, Armonk, NY, USA). Comparison between the results of the research groups was carried out by one-way analysis of variance (ANOVA) followed by “Tuckey” post-hoc test to report the statistical significance. *p*-values < 0.05 were considered as an indication of statistical significance.

## 5. Conclusions

We demonstrated that the induction of apoptosis and cell cycle arrest are involved in the anti-proliferative effects of n-3 PUFAs in vivo and in vitro. Our study examined the preventive role of n-3 PUFAs on HCC production and progression induced by STAT3 signaling. Therefore, it is strongly proposed that n-3 PUFAs could be used as adjuvants for cancer treatments by targeting STAT3 and its downstream signaling proteins, which might be therapeutic targets with a good potential for HCC treatment due to its implication in cell growth and survival. However, appropriate clinical trials should be carried out before directly transposing to humans. Further studies are needed to elucidate this aspect by utilizing other specific apoptosis agents and comparing their actions with those of other drugs in inducing apoptosis and the mechanisms involved.

## Figures and Tables

**Figure 1 molecules-27-03032-f001:**
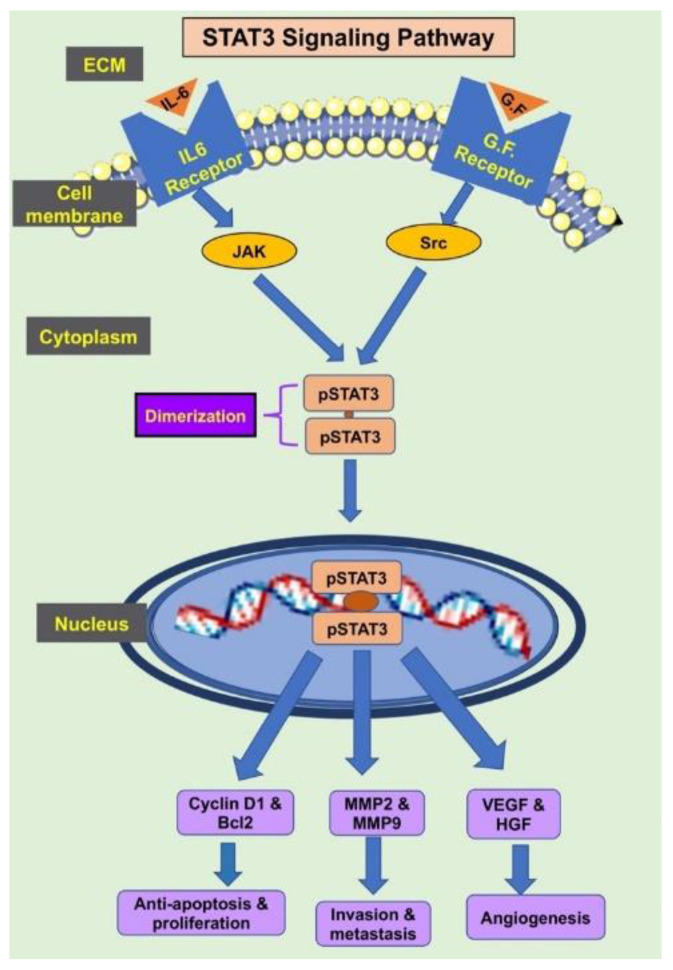
Activation of STAT3 induces the transcription of the target genes that promote cell growth, anti-apoptosis, migration, invasion, metastasis, and angiogenesis.

**Figure 2 molecules-27-03032-f002:**
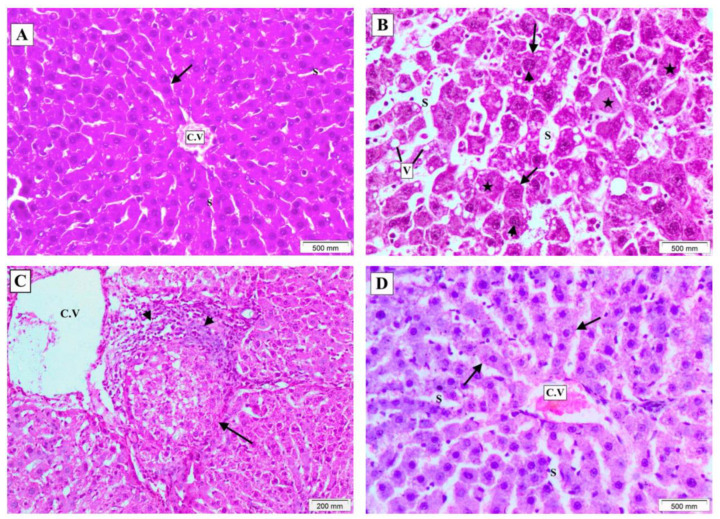
H&E-stained sections showing histopathological changes in liver tissue of all research groups. (**A**) Control group showing normal hepatic architecture formed of cords of acidophilic hepatocytes with uniform nuclei (arrow) radiating from the central vein (C.V), separated by blood sinusoids (S). (**B**) DEN group showing lost architecture with focal necrosis and hyalinosis (stars), atypical pleomorphic hepatocytes (arrows) with pleomorphic hyperchromatic nuclei and distinct nucleoli (arrowheads), markedly vacuolated cytoplasm (V), and dilated sinusoids (S). (**C**) DEN group showing large nodule with central necrosis (arrow), marked inflammatory infiltrates (arrowheads), and dilated central vein (C.V). A pseudo-glandular pattern can be noticed (rectangle). (**D**) n-3 PUFAs group showing nearly normal liver architecture with regular hepatocytes’ cords except for mildly dilated sinusoids (S) and congested central vein (C.V). (magnification A,B,D × 200, C × 100).

**Figure 3 molecules-27-03032-f003:**
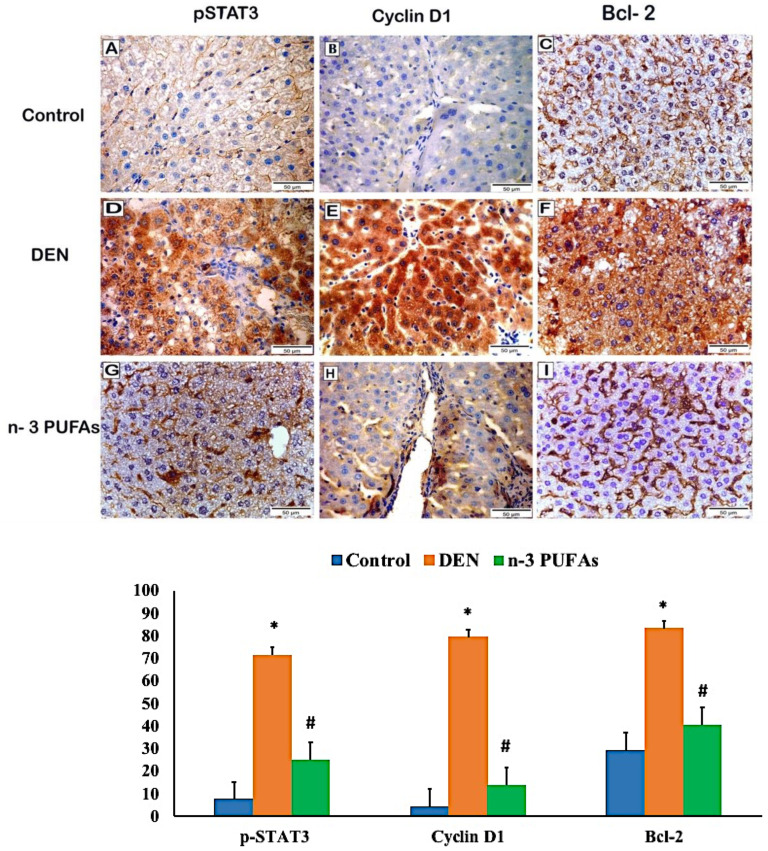
Immunohistochemical staining for p-STAT3, Cyclin D1, and Bcl-2 antibodies in different research groups, magnification 400, scale bar = 50 μm. (**A**–**C**) Control group showing a positive reaction for p-STAT3, cyclin D1, and Bcl-2 antibodies. (**D**–**F**) DEN group showing strong positive expression of all antibodies. (**G**–**I)** n-3 PUFAs group showing regression of immune reactions for all antibodies. Outcomes are expressed as mean ± SD; one-way ANOVA followed by post hoc Tukey’s test for intergroup comparison. * significant from the control group, # significant from the DEN group, *p* < 0.001.

**Figure 4 molecules-27-03032-f004:**
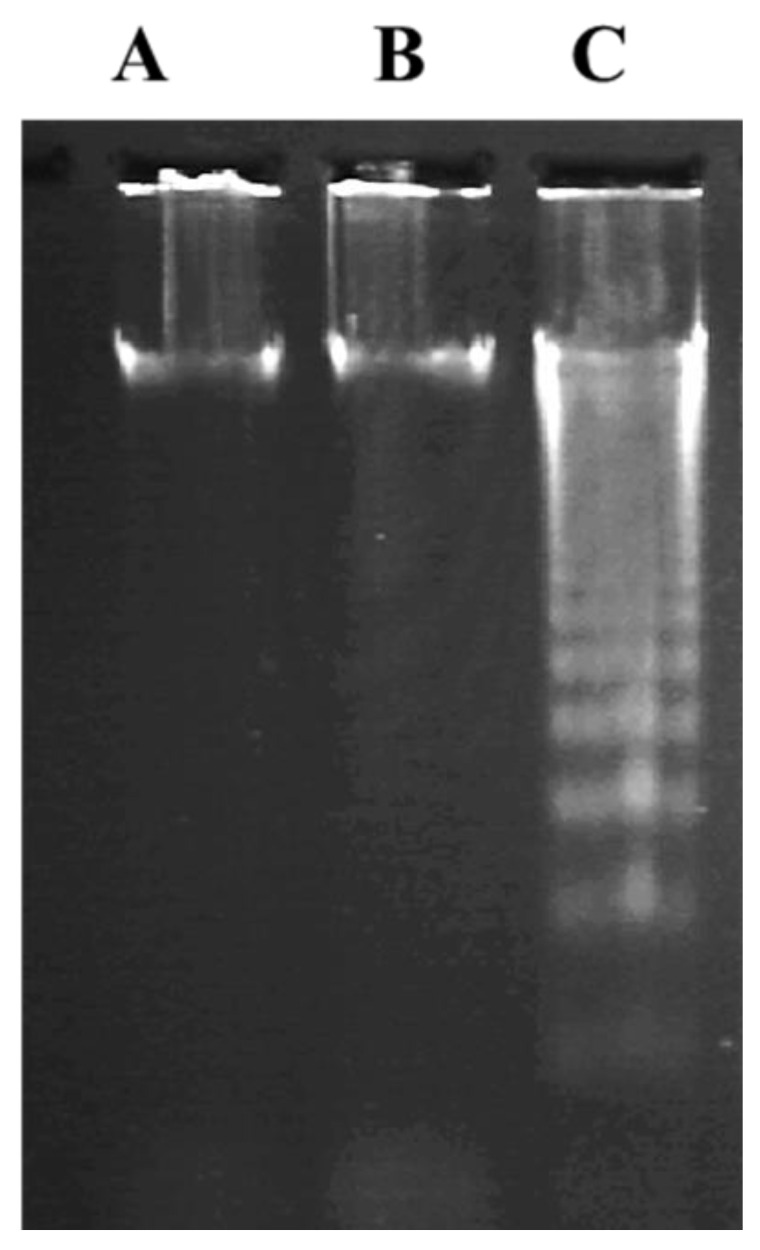
Influence of n-3 PUFAs on DEN-induced hepatic DNA. Lane (**A**): no DNA fragmentation in normal control. Lane (**B**): no DNA fragmentation in DEN-treated HCC rats. Lane (**C**): weak DNA fragmentation in HCC rats treated with n-3 PUFAs.

**Figure 5 molecules-27-03032-f005:**
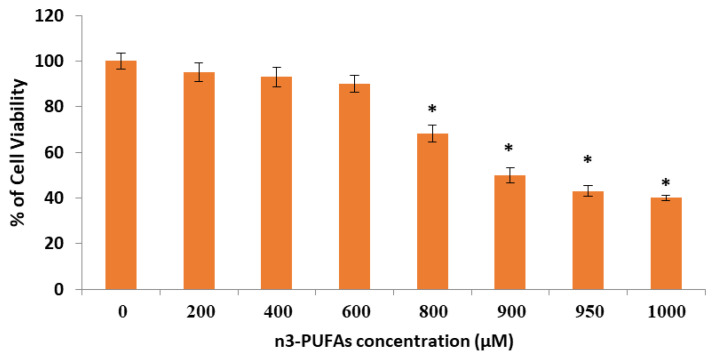
Effect of n-3 PUFAs on the viability of HepG2 cells measured using 3-(4,5-dimethylthiazol-2-yl)-2,5-diphenyl tetrazolium bromide assay. Data were from at least three separate experiments. IC_50_ for HepG2 cell line at 24 h was 900 μM. Cell viability in control groups were 100% * *p* < 0.05, compared with the control (0 μM) group.

**Figure 6 molecules-27-03032-f006:**
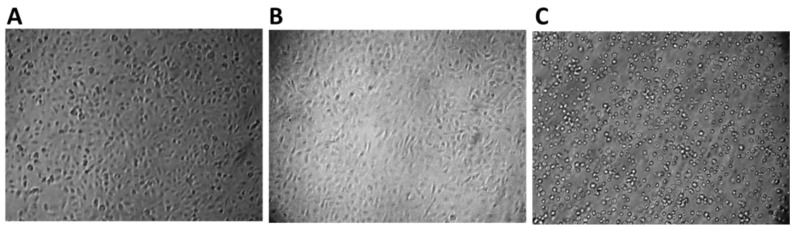
Morphological changes of HepG2 cells at 40 × 10 magnifications. (**A**) Normal HepG2 cells. (**B**) Cells treated with 450 μM (IC_25_) n-3 PUFAs, only slight morphological changes were present. (**C**) Cells treated with 900 μM (IC_50_) n-3 PUFAs are detached, and the numbers of apoptotic bodies and cell shrinkage increased. The results are from one representative experiment of the three independently performed experiments that showed similar patterns.

**Figure 7 molecules-27-03032-f007:**
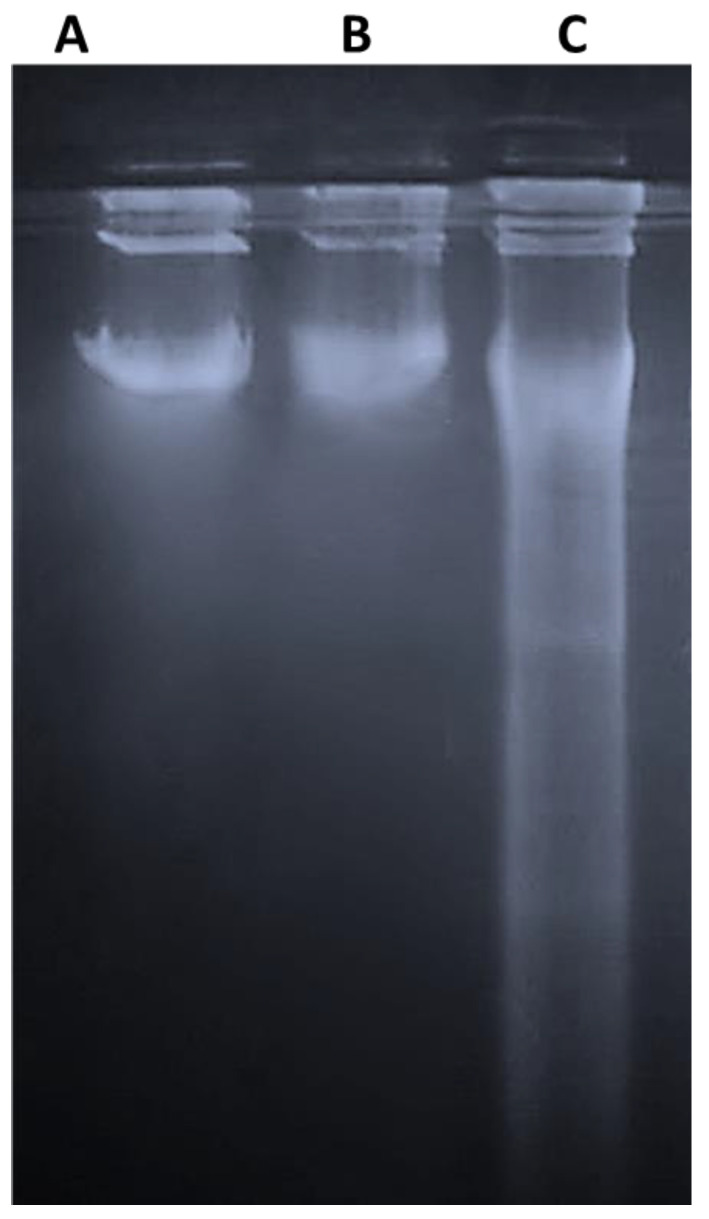
Detection of DNA fragmentation by agarose gel electrophoresis. (**A**) untreated cells; (0 μ M), positive control HepG2. (**B**) cells were exposed to n-3 PUFAs at half IC_50_ concentration (450 μM) for 48 h; (**C**) DNA fragmentation at IC_50_ (900 μM). Three experiments were performed with similar results.

**Figure 8 molecules-27-03032-f008:**
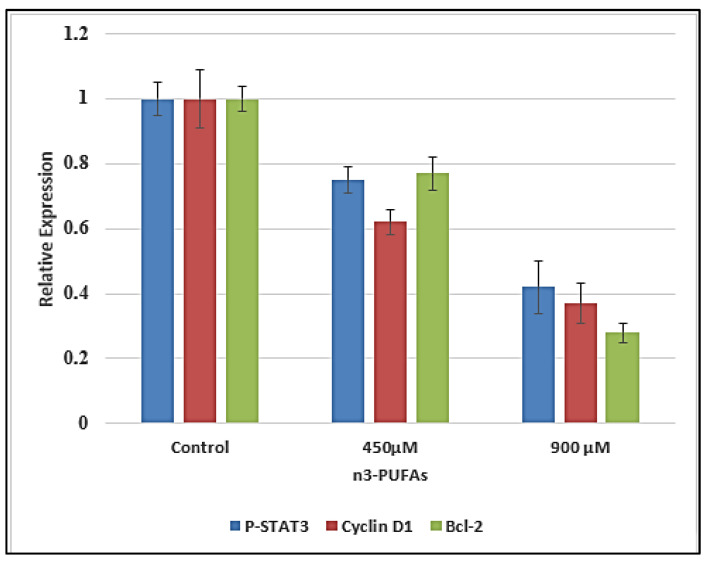
Gene expression in HepG2 cells after n-3 PUFAs treatment of 24 h. Outcomes are expressed as mean ± SD.

**Figure 9 molecules-27-03032-f009:**
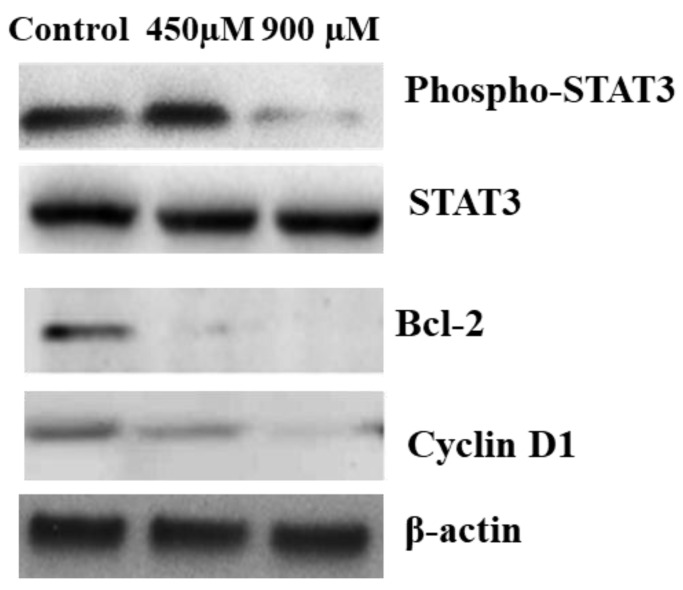
Western blot analysis of p-STAT3, STAT3, Bcl-2, Cyclin D1, and β-actin protein expression in HepG2 after 24 h n-3 PUFAs treatment.

**Figure 10 molecules-27-03032-f010:**
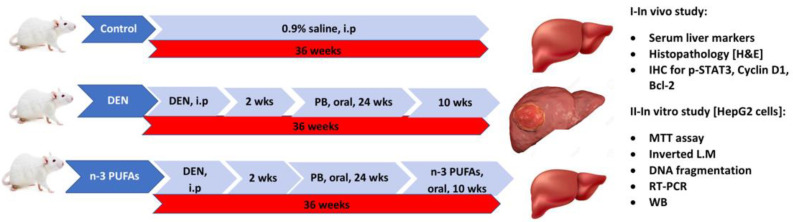
The timeline of the study.

**Table 1 molecules-27-03032-t001:** Effect of n-3 PUFAs on serum liver markers in the experimental groups of rats.

Group	ALT (U/L)	AST (U/L)	ALP (U/L)	GGT (U/L)	Albumin (g/dL)
Control	55 ± 1.2	98.4 ± 3.5	307 ± 5.7	56.7 ± 5.6	5.2 ± 1.3
DEN	206 ± 6.7 *	264.5 ± 5.7 *	311 ± 10.6	60 ± 9.4	4.8 ± 0.98
n-3 PUFAs	124 ± 5.6 #	194 ± 4.8 #	298.5 ± 6.7	49.3 ± 7.4	4.23 ± 0.64

* significant to control group, # significant to DEN group at *p* < 0.05.

## Data Availability

All the data obtained during the study are readily available from the corresponding authors upon reasonable request.

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
