# Peer review of "Omega-3 Polyunsaturated Fatty Acids Provoke Apoptosis in Hepatocellular Carcinoma through Knocking Down the STAT3 Activated Signaling Pathway: In Vivo and In Vitro Study"

_molecules, 2022, doi:10.3390/molecules27093032_

Round 1

Reviewer 1 Report

This is very interesting manuscript which shows that Omega-3 polyunsaturated fatty acids that are not synthesized in human body and must be for proper human body function provided with diet, could induce apoptosis and also have impact on cell signal transduction leading blockage in Stat3 synthesis. It is possible that like vitamin D, also other factors like  Omega-3 polyunsaturated fatty acids could be involved in regulation of gene expression. In this contex  Omega-3 polyunsaturated fatty acids could have, beside their known function that increase immune system, also could be involved in gene expression regulation. As presented in this manuscript  Omega-3 polyunsaturated fatty acids  could decrease expression of STAT3, a very popular factor which expression usually increase in several cancers. In this aspect authors should also check the effect of different doses. 

The topic of manuscript is very interesting, but in the group III from my knowledge there is to high dose of  Omega-3 polyunsaturated fatty acids and this dose on Fig. 4 form standard DNA lader. So this manuscript looks like very big work done by scientists, but I am affrait that it is not possible to use safetly dose like 400mg per kg/day. I suplement  Omega-3 polyunsaturated fatty acids as prophylactic and I take  Omega-3 polyunsaturated fatty acids ALA in one softgel include from 160-240mg. I take this amount for ones. There are together 800mg of mixture of different polyunsaturated fatty acids This is for health protection. When we use quercetin in lower dosage could also stimulated immune system, but in higher concentration also induce cell death, mainly by apoptosis. 

At first we have to choose the right dosage that will be active, but will not kill. Even if usually  Omega-3 polyunsaturated fatty acids are connected with health, in high concentration could be danger, because of their solubidity in fat. We can do such experiments in rats, but it will be rely difficult to interpolate such studies into human, because of dosage, to high could be toxic for liver.

For me there is one lesson from this manuscript. It looks that everything what we eat, drugs and food constituents could imply on cell signalling and signal transduction by regulation for example gene expression, like  Omega-3 polyunsaturated fatty acids , as well as vitamin D. Even if we will ad a high amount of some agent we have to start our research from thinking if this dosage is a normal, and we can interpolate this into human. On Fig.7 there is a rely good example that in results you have apoptosis with necrosis. 

I must write that this research results rely looks good, but from my point of view you should start look if your studies will be important to transfer in the future for humans. 

Reviewer 2 Report

Manuscript Number: Molecules-1690383

In this article entitled “Omega-3 Polyinsaturated Fatty Acids Provoke Apoptosis in Hepatoclellular Carcinoma Through Knocking Down the STAT3 Activated Signaling Pathway: In Vivo and In Vitro Study”, the authors studied the effect of omega-3 on hepatocellular carcinoma (HCC) via enhancing apoptosis.

The article is well written and the methodology is good. However, some points should be taken into account before its acceptance.

Comments to the authors:

  • Line 52, what “developing” refer to? Is it developing countries, because the authors cited Africa and Eastern Asia. Please clarify.
  • “EC matrix”, in line 58, should be “ECM”, which stands for extracellular matrix.
  • Lines 64-68, the two sentences report almost same things. It should be better to fuse them-up in a single sentence. Otherwise, the readers will notice a kind of repetition.
  • In figure 1, i) it would be better to give a different color for the nucleus. ii) Why do the authors are reporting 2 separate boxes for “pSTAT3”? Normally it’s the same, which bound with either JAK or Src. iii) the nuclear envelope sounds thinner and has different aspect while compared to the cell membrane.
  • Line 73, introduce the abbreviation for JAK.
  • - It sounds that the authors prepared the first draft of the manuscript with material and methods before the results and discussion. But following the journal requirements, in which the latter is followed by the material and methods, the authors forgot to arrange some items, particularly for the abbreviations. Please check.
  • The statistical analyses need to be checked specifically for the figure 5. In fact, some other doses (other than 950 and 1000 µM) may have significant effect on the percentage of viability.
  • - There is a problem with the scale bar of the histological slides, specifically figure 2.
  • English language is fine but some minor mistakes should be revised, particularly the punctuation.

Round 2

Reviewer 1 Report

This is very important manuscript. More detailed inspection revealed that from 800 to 1000 µM could be important for 3 PUFA in vitro activity for HepG2 cell line.  Therefore it is a good idea to transfer IC50 (900 µM) into proper curative dose for humans. The most important is choosing a curative dose for humans. Please analyse why sometimes (Fig.4) PUFA caused apoptosis (apoptotic lader is a confirmation), while on Fig.7 the same agent in ½ dose (450 µM) induced apoptosis and necrosis (different lenght of DNA form characteristic morfology like seen on Fig.4; smear).

A very important are results showing that diethyl-nitrosamine, nitrosamines are constituents of cigarettes smoke, could induce expression of STAT3, cyclin D related to cell cycle (first connected with cells signal transduction, second with cell cycle; both over-expressed in many cancers). Moreover, simultaneously was also coexisted with high expression of Bcl-2, factor that inhibit apoptosis. As we can see in this manuscript unhealthy cigarettes smoke constituents are able simultaneously imply on several different cellular pathways, at the same time.

I can recommend this manuscript for publication

Reviewer 2 Report

The authors ameliorated the manuscript according to the addressed comments. Nevertheless, some minor comments should be taken into account before publication of the manuscript.

Comments to the authors:

  • Change in the authorship should be justified,
  • Figure 2: i) the scale bar still is not correct in figure 2 and not readable for Fig. 2C. Furthermore, magnification for figure 2A can never be × I think it’s either ×200 or ×250 only. Likewise, magnification for Fig. 2C may be only ×100 and its corresponding scale bar is unreadable. ii) Details about the magnifications and scale bars should be given once for all the slides at the end of the figure 2 legend, no need to be repeated several times.
  • Figure 10 is labeled on the top and also in the bottom as figure legend.
